# The Clash of Cultures of Radical Enlightenment and Humanism Open to Transcendence. The Perspective of Pope Benedict XVI

Janusz Węgrzecki

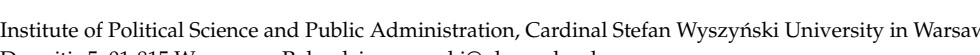

Institute of Political Science and Public Administration, Cardinal Stefan Wyszyński University in Warsaw, Dewajtis 5, 01-815 Warszawa, Poland; j.wegrzecki@uksw.edu.pl

**Abstract:** The article analyzes the content of the Pope's speeches discussing, reconstructing and interpreting the concept of two dominant western cultures and their mutual relationships to the perspective of Pope Benedict XVI, who calls them the culture of radical enlightenment and the culture of humanism that is open to transcendence. The article identifies fundamental contentious issues including: anthropological issues, human dignity, political anthropology, freedom, reason, its rationality, and the role of religion in the public sphere. Thus, the article provides a positive answer to the question of whether the perspective of the clash of cultures outlined by Samuel Huntington can be cognitively used in interpreting the contrast of cultures presented from the perspective of Pope Benedict XVI. However, contrary to Huntington, who describes the clash of western cultures with other, non-western cultures, Pope Benedict XVI claims that there is a clash between two dominant western cultures.

**Keywords:** clash of culture; enlightenment; Christianity; dignity; freedom; reason; Pope Benedict XVI

## 1. Introduction

Of numerous contentious issues, the most fundamental one seems to be the dispute over culture, perceived as a whole, and thus synonymous with civilization. What we can observe, not only in Europe or North America, is the clash of cultures. On one hand, we have the western culture (Roszkowski 2019), or humanism open to transcendence, as Pope Benedict XVI called it in Paris, 2008 (Benedict XVI 2017). On the other hand, we have a culture of radical enlightenment (Ratzinger 2010).

Benedict XVI's approach is highly original. The paper interprets the thought a former pope who left office in 2013, and the current pope is Francis. It ranges widely in understanding the relationship between religion and politics or culture. The Pope's approach differs from political theology, both from the theological (Metz 1997; Scattola 2007) and the political philosophy point of view (Schmitt 2000). According to the Pope, religion should keep its distance from current political life. Religion should remain on a meta-political level and should not directly involve itself in politics (Milbank 2006). It differs from the history of particular ideas such as (Brague 2005), being human (Brague 2015; Scruton 2017), or even from a detailed analysis of contemporary religious problems (Besancon 2017) despite the fact that it ponders specific problems such as the relationship between Christians and Jews (Benedict XVI 2019) or Catholic priesthood (Benedykt XVI 2020, pp. 21–57) or the role of the pope (Weigel 2020). While considering the synthesis of modernity, the Pope's approach comes close to the political philosophy that interprets modernity as the contemporary version of gnosis (Voegelin 1992), rival versions of moral traditions (MacIntyre 2007), the secular age (Taylor 2007), disputes about natural law (Finnis 2011a), and religion in the public sphere (Rawls 1993; Habermas 2005; Finnis 2011b).

The article attempts to find an answer to the following research questions: How does Pope Benedict XVI conceptualize and interpret the culture of radical enlightenment? How does he perceive the culture of humanism that is open to transcendence? What fundamental

contentious issues can be found in the clash of both cultures? Can the perspective of the clash of cultures, as outlined by Samuel Huntington in 1996, be cognitively used in interpreting the contrast between the cultures presented from Pope Benedict XVI's point of view? (Huntington 2011) In order to find the answers to the first two questions, we need to reconstruct the content of Pope Benedict XVI writing. The analysis of the Pope's speeches allow us to formulate a hypothesis that the fundamental contentious issues include: anthropological issues, human dignity, political anthropology, freedom, reason and its rationality, and the role of religion in the public sphere. Similar verification will be applied to the hypothesis concerning the potential application of Huntington's concept to interpret the Pope's point of view, which can be boiled down to his conviction that we are witnessing a clash of cultures. In this paper, "civilization" has the meaning "culture". Contrary to Huntington, who believes that western and non-western cultures clash, Benedict XVI sees the clash occurring between two dominant western cultures.

The article discusses the clash of cultures within the above-mentioned contentious issues. Firstly, it discusses, analyzes, presents, and interprets the characteristics of both cultures. Simultaneously, it presents the clash of cultures in light of the issues creating the tension, as they are perceived differently by both cultures. It concludes with reference to a wider context of the clash of cultures, from the point of view of Pope Benedict XVI, and formulates identifiable implications of the discussed process.

## 2. The Culture of Radical Enlightenment

Let us begin by presenting a picture of the world painted by the culture of radical enlightenment. As far as the content is concerned, we can express a few fundamental convictions, which constitute a particular version of anthropology. One vision is that of a human being as an individual, a person who does not develop any significant ties or at least someone who can function in near isolation. An individual operating by negative freedom enables them to attain positive freedom, which is consistent with the enlightenment canon. A human being is perceived firstly as an almighty demiurge, totally in control of their fate and the surrounding world. He achieves this by following scientific rationalism and science-related technology. At the same time, this is an individual who believes that neither God nor morality, understood traditionally, exist. In public life, he follows principles of negative secularism. There is no place for God or religion in public areas, as they are confined to an individual human mind. A man of radical enlightenment supports new morality, which positively evaluates our behavior by calculating the consequences. The moral evaluation of an action is determined by its acceptance or rejection. He believes that there is not only scientific and technological progress, but also moral progress, which is identified within the content of new morality. Moreover, such a person believes in the validity of political moralism and accepts wide-range actions which facilitate progress. Such actions must be political in nature, thus forcing the state to become involved. In the name of progress, he accepts that there might be victims—other people. He thinks in line with the political utopia that the world inevitably approaches, which directs political actions and justifies them, acknowledging them to be valid. This utopia adopts the form of a more or less specified model of individual and collective life. Following the rules of political moralism and a specific utopia, a human being develops tools which ensure the dominance of the *creed* typical of radical enlightenment, and thus removing all competitors from the public stage by treating them as usurpers.[1] According to Benedict XVI, the dispute with the culture of radical enlightenment concerns several fundamental problems: human dignity, political anthropology, freedom, the rationality of reason, the role of religion and the legitimacy of the question about God. The anthropological issue is undoubtedly one of the most topical social issues. The dispute concerns the appropriate vision of a human being. The search for the ultimate truth about humans leads to anthropology based on dignity, which is founded on a conviction that every human being possesses dignity, both the creator and the person subjected to the act of creation. This anthropology view leads to a society created as a set of personal connections (Węgrzecki 2020, pp. 116–20).

Dignity is interpreted as something internal, found, prior, irremovable, permanent, and outstanding in the world of nature. Dignity belongs to a human being intrinsically. It does not depend on someone else's will. It is not given by anyone else or by man himself. Since dignity does not depend on will, it cannot be taken away from us. A human being is born with dignity and does not lose it until his death. Dignity demonstrates that a human being possesses something outstanding and specific in comparison to natural beings. One might say that dignity reflects "humanity". The culture of radical enlightenment tends to overlook this fundamental difference. Benedict XVI claims that radical advocates of philosophical enlightenment believe that "a human being should not believe that he is someone different from other living beings, and therefore should be treated as they are treated" (Ratzinger 2010, p. 260). Benedict XVI's approach to human dignity contrasts with Kant's position. Kant opened a route for three false exegeses (from the Pope's perspective) in the 20th century: the relationship between the Church and Jesus leads to limited moral dimension of Christianity, human freedom leads to a relativistic position between the will of man and cultural moral rules, and religion is restricted to private sphere only (Kant 1954).

Of the many descriptions of dignity, let us quote the following one: "Cultural differences obviously exist; however, they do not prove heterogeneity of the human kind, but, on the contrary, they prove our specific nature and difference from the animal world. Which animals can develop abstract concepts and linguistic forms, beliefs and moral norms, customs and conscience? It is *potentiality* of a human being, able to develop it all, that determines our nature! (...) Equally detrimental results were produced by attempts at depriving humans the social feature of our nature" (Roszkowski 2019, pp. 170–71). The quote, although a bit phenomenological, provides an apt description of the specificity of "humanity", which, when compared with the surrounding non-human world, becomes dignity. What are specifically human are our epistemic abilities, creating concepts, using language, self-knowledge, including moral knowledge (conscience), and establishing connections with similar beings (other people). Roger Scruton distinguishes self-consciousness from consciousness, which at least some animals possess. Self-consciousness, as an exclusively human feature, allows us to see our face as "me" and enter into relations with others as "me–you" (Scruton 2017). A human being is oriented toward creating societies through communities.

Human specificity and dignity are reflected in political anthropology. As Chantal Delsol claims, the interpretational choice of dignity anthropology was made at the very beginning of Europe. This dignity perception of a human being is currently being questioned. Dignity is no longer understood as something internal, but is becoming something external. Dignity is thus determined by external will, and this leads to some limitation in the universality of dignity. It is now someone's will that determines who should be given dignity and who should be denied it. Dignity is no longer a universal feature. Some people are denied it. This is clearly seen in eugenics. Delsol claims that: "Both in totalitarian eugenics and liberal eugenics there is always somebody (an ideological group, state or individual) who believes they are authorized to decide whether another human being is a person or not (...) dignity is no longer an inherent feature of every single individual, but it is given, granted, licensed" (Delsol 2018, pp. 58–59).

Political anthropology is currently an area in which there is constant discourse over different interpretations of a human being. One of them, containing the dignity line of reasoning, is called limited anthropology, and the other, unlimited anthropology. Gierycz and Mazurkiewicz offer an interpretation of both anthropologies. "Limited anthropology should in principle take into account the double limitation of a man. The first one is a consequence of the original sin. Everything a human being gets down to doing is tainted with imperfection (...) The second limitation is of more fundamental nature. It does not stem from *corruption* of human nature, but from the fact of being a creature. Unlimited anthropology occupies the opposite positions, claiming that a human being is perfect in moral and epistemic aspects" (Gierycz and Piotr 2015, pp. 20–21).

References to both political anthropological interpretations can be found in EU public policies, with a visible shift towards the unlimited anthropology preference. This is clearly noticeable in the area of financing particular EU policies (Gierycz 2021, pp. 385–447).

Another dispute concerns our understanding of human freedom. The culture of radical enlightenment depicts human beings as free and almighty. It emphasizes negative freedom so that a person could do what they want, without encountering any external obstacles or barriers. One potential barrier can be found in classical values of western civilization, such as truth, goodness, and beauty. In the culture of radical enlightenment, these are presented as rivals to freedom. Particular controversies are aroused by the truth about goodness, the moral truth which guides human conduct. Another rival to freedom is God and religion. In addition, all external thoughts, namely tradition, are also rivals to freedom. They are all tied to family and national bonds which come together to form cultural identity. Deneen believes that a relationship between freedom and threats to it in the shape of time, place, and nature are of fundamental significance for a particular culture (Deneen 2018). Freedom is threatened by the past and the future. Only the present guarantees freedom. Another threat to freedom is our attachment to place. A free human being, the type of person that the culture of radical enlightenment wants to embody, lives unattached to any place, that is, they are everywhere and nowhere. Our bonds with a particular area, such as our family home, region or national bonds, limit us. Consequently, we must weaken patriotism and give priority to supranational structures. A free human being is cosmopolitan. "No place" is their homeland. Nature turns out to be another threat to freedom. Progress demands liberation from all natural conditions, especially from natural moral law.

A person who is not constrained by time, place and nature becomes almighty. Free, and thus unlimited action is based on technological possibilities. However, people are not indifferent to these technical possibilities. According to Benedict XVI, what are worrying are the "possibilities of self-manipulation acquired by people" (Ratzinger 2010, p. 251). They become cognizant of everything that is made possible by technology as some sort of mental pressure or even absolute constraint. Since we are free and can clone, because we are in possession of cloning technique, therefore, feeling forced to act, we clone. Similarly, being free and knowing how to use a human body as a spare parts store, we create them. Alternatively, having mastered the technique of building an atomic bomb, we proceeded to mass produce it. A free person, who can use terrorism as a technique to cause mass hysteria and destruction of life, performs acts of terrorism. A man who declares themselves to be absolutely free, and thus almighty, instead turns out to be submissive, acting out of necessity as described by positivism or under strong mental and psychic compulsion to use all available technological solutions.

Another contentious issue is the rationality of reason. According to Benedict XVI, radical enlightenment is not as rational as it would like to be perceived. There are two limitations to rationality. On one hand, it refers only to a fragment of truth, as it narrows cognition to positivistic empirical science. On the other hand, it limits cognition only to possible meanings and convictions of an ideological nature. Ratzinger/Benedict XVI states: "Tangled ideology of freedom leads to dogmatism, which turns out to be increasingly hostile to freedom" (Ibid, p. 257). Contrary to its claims, radical enlightenment cannot become a universal culture. Linked to it, there are still great cultures based on the traditions of religious and moral thinking. The future lies not in driving other cultures away, but in dialog, and in the openness of radical enlightenment to other cultures, while the latter must acknowledge valuable elements of enlightenment. Benedict XVI admits that dialog between cultures is not only possible, but necessary. Dialog is accompanied by reflection on our own culture. According to him, "it is necessary for both parties to reflect on themselves and to be ready to improve" (Ibid, p. 262).

The dispute also concerns the attitude to religion. Great cultures have always been close to religious traditions and questions about God. Religions should be considered a culture-forming factor. Christianity brought sensitivity to values, including sensitivity to

freedom. God is free and a human being is free. Radical enlightenment originated within a culture rooted in Christianity. Nevertheless, it wants to break all its ties with Christianity. It considers itself self-sufficient and complete. It is unaware of its internal contradiction which were presented above. Radical enlightenment is directed by negative agnosticism. We live and organize our public, social, political, and legal life as if God did not exist, and even if God exists, this is of no practical significance to us. Consequently, we treat religious institutions only as private associations of the faithful, non-governmental organizations, organizations typical of a civilized society. In the political process, we agree with their position, their right to operate, we define norms and limits of their existence in the public sphere. According to Benedict XVI, this is not enough, as religion brings ethics to public life, and ethics are necessary for the public sphere and the state to function (Węgrzecki 2019).

According to Benedict XV, I we must oppose "all forms of hostility to religion, which limit the public role of the faithful in civil and political life" (Benedict XVI 2011, n. 8). The situation in western democratic countries is also worrying. "There are (...) more sophisticated forms of hostility towards religion, which are sometimes manifested in western countries by denying history and religious symbols that reflect the identity and culture of most citizens. They often fuel hate and prejudice and are not consistent with the peaceful and balanced vision of pluralism and secularity of institutions. Moreover, there is a risk that new generations will never encounter invaluable spiritual heritage of their countries" (Ibid, n. 13).

Radical enlightenment preserves some moral concepts, such as justice, peace, or care for the environment. Old names remain, but the content is new. Instead of goodness, there are calculations of consequences. No conduct is internally good or bad. We cannot evaluate it from the point of view of goodness and evil. What really matters for the evaluation is an anticipated effect, a consequence. If the effect is acceptable, then the action leading to it is acceptable. Apart from the evaluation of the effects of actions, the ethics of consequences contain one more element. An action that is possible thanks to technology is closely related to the positivistic model of empirical science, and along with it comes the already-mentioned mental and psychological compulsion. Anticipation of positive effects and the mechanism of compulsion force us to act. The specificity of the ethics of consequences is well illustrated by an action referring to social peace. Anti-establishment movements use violence, destroying cities, shops, and banks. They consider their actions appropriate and justified by the positive evaluation of their consequences. The ethics of radical enlightenment are not only extremely subjective and characterized by relativism, but it poses a threat to democracy. Benedict XVI states: "The spread of foggy cultural relativism and utilitarian and hedonistic individualism weakens democracy and is conducive to dominance of tycoons." (Benedict XVI 2010).

New morality is inconsiderate to human dignity. Acknowledged positive consequences of actions may demand sacrifices. The dignity of every person is not an absolute value. New morality as public ethics is being transformed into political morality. Radical enlightenment preaches scientific, technological, and moral progress. However, moral progress takes the form of political morality, which puts "political utopia over dignity of an individual" (Ratzinger 2010, p. 253). The expected changes can only be implemented by political authorities. Thus, power becomes the weapon of progress. Those in power determine the direction of progress. Politicization comprises both an ideological component and the consequence of actions based on science and technology. Those who wield power dominate the area of public life. Radical enlightenment has dominated the political public sphere and the whole legislative process. Both negative and positive freedom are politicized. On one hand, there are no obstacles to oppose acting in line with subjective expectation. On the other hand, the accomplishment of what is expected and desired requires positive freedom. Politicization means limiting the objections raised by others against the fulfillment of subjective expectations. Effectiveness may be ensured by giving this subjective expectation a group feature. It is no longer an individual who wants to act in this way or another, but the whole group and the state is supposed to allow this action,

thus making all social objection to it ineffective. Political morality makes a certain form of positive freedom publicly valid, forcing everybody to acknowledge this specific form of positive freedom as a norm of social life.

Political morality is a mechanism leading to an ideal, expected state, that is, the height of progress. Radical enlightenment adopts the utopia perspective in its thinking. What is new is always better. Progress in science and technology does not inspire controversies. Is there parallel moral progress? Benedict XVI states: "the growth of our capabilities is not accompanied by the development of our moral power. Moral power has not grown along with the progress of science; on the contrary: it diminished because technical mentality narrows morality down to the subjective sphere" (Ibid, p. 252).

### 3. The Aporias of Radical Enlightenment

The first one concerns the concept of freedom. On one hand, radical enlightenment preaches, promotes, and expands freedom, but on the other hand, it limits freedom. Imposed on everyone, freedom "to", consistent with the content of radical enlightenment, limits the freedom of others, those who do not share the enlightenment creed.

Another aporia refers to understanding what is rational. Radical enlightenment refers to scientific rationality, which combines mathematics and empiricism. Such rationality narrows cognition. It puts the knowledge of the other kind beyond the scope of rational, true cognition. It rejects ethical, philosophical, and theological rationality that has been developed over centuries. All meanings of intellectual and spiritual cognition are beyond its reach, as they do not fit in with this scheme of rationality. In this concept of science, in natural, social and human science a portrait of a human being is created, deprived of soul and free will, being a body that is temporarily granted reason. Material genealogy of human kind is created. In the beginning, there is no-reason, irrationalism, nonsense. As a result of some unspecified processes, by mere coincidence, no-reason transforms into reason, and then it moves on to the no-reason stage again. This picture evokes a question about freedom. In the beginning, there is no freedom, but does it appear later on? For many people who acknowledge the existence of only this kind of rationality, a human being is never free. Freedom does not exist, there are only necessities to which we are passively submitted, even if we are temporarily given reason. Radical enlightenment preaches freedom, but at the same time, it claims that rationality is exclusively limited to science rooted in empiricism. This science, however, boldly claims that freedom does not exist. By acknowledging the rightness of the rationality of positivism, radical enlightenment then contradicts freedom. Enlightenment is not such an advocate for freedom as it would like to be and as it declares itself to be.

The next aporia consists of combining scientific positivistic rationalism and specific ideology. Did enlightenment manage to combine something that seemed to be impossible to join, and thus did it obtain a synthesis of two opposite perspectives of rationality? Such synthesis only seems successful. Scientific empirical rationality excludes rational claims and beliefs that makeup enlightenment ideology. The enlightenment creed contains two contradictory rationalities: that of empirical science—such a model of science is widely accepted—and that of ideology. According to the former, there is only one rationality, represented by science. Everything else is not rational. If this is so, then many claims and convictions of radical enlightenment are irrational, including, above all, the concept of negative freedom. It cannot be defended on the grounds of the positivistic concept of science. How can we announce something to be rational—in this case, negative freedom—when from the perspective of another accepted element—empirical science—it is considered irrational?

The presence of such diverse elements: empirical science and ideology lead to another aporia. Ideology claims that a human being is an almighty and free entity who can do whatever he desires. On the other hand, scientific empiricism radically limits human freedom. Almighty or passive, acting freely or as a result of some necessity? The rationality

of positivistic science is connected with technology. Science leads to practice, which consists of freedom-less fulfillment of particular technological procedures.

## 4. The Culture of Humanism Open to Transcendence

Let us take a close look at the Christian culture, which became the foundation of Europe and can be defined as humanism open to transcendence (Benedict XVI 2017). This culture is universal, therefore, Benedict XVI treats the term "Christian culture" with reserve, as it might be misunderstood as a confession of faith. It is not a culture created by Christians exclusively for Christians. Admittedly, one of its major sources is Christianity. This culture evokes what is permanent, fundamental, and elementary, and this is a particular vision of a human being; personalism. This anthropology is based on the category of dignity, ethics based on natural law and objective values of truth, goodness, beauty, justice, and freedom. Therefore, it is better to describe it as humanism, which concerns all people, believers, and non-believers. Humanism that manifests dignity, anthropology that opens the human world to the ultimate, including the question of God. Humanism which does not determine the existence of God, but is open to the existence of transcendence and the possible dialog with the Supreme Being.

The fundamental feature of the culture of humanism open to transcendence is that it seeks God in the form of existential eschatology. It seeks what is original, fundamental, ultimate, unchangeable. It goes beyond what is transient, not stopping at what is directly given to us, but moving on to what is ultimate, or eschatological. The main element of this culture consists of seeking the deepest features, going beyond unearthliness. We might differ in how we identify the deeper reality, but our common trait is the search for it.

Eschatological search for the fundamental and ultimate issues is based on reason. This reason is rational. Its action is prior to specific cognition, which constitutes only its part. It excels and goes beyond all technical cognition. Nevertheless, in order to act properly, it needs appropriate tools, and these are the secular sciences. The primary ones include grammar and logic, philosophy and theology, as well as law and medicine. Then, there is the wider spectrum of the humanities and social sciences, as well as natural sciences. The latter demonstrates the mathematical structure of the empirical world. The search for the fundamental will also include an answer to the question of why what is defined as empirical is subjected to the rules of mathematics. Natural sciences do not ask this question. They just function in such obviousness. However, this question is eschatological in nature as it concerns seeking what is ultimate, thus going beyond what is transient and empirical.

The search for the ultimate, going beyond the transient, is not conducted individually, but in a community. It is in the community that this search becomes fruitful. Experience, learning and cooperation within one generation and between generations may only take place in a society, and it is absolutely necessary to accomplish this universal aspiration. What is fundamental is captured as a vision of God and a human being and the world of values. Seeking God, human dignity and objective values of truth, goodness, and beauty requires freedom, and freedom is accomplished within social ties. A man is free only within the experienced social ties, within the society defined as community. Society is a community of free people.

The true sense of freedom calls for personal ties and ties with other values. There is a unity of truth, goodness, beauty, freedom, and justice. Freedom deprived of social ties and ties with other values is distorted.

Having outlined dignity anthropology, let us move on to other features of the culture of humanism open to transcendence. It is a culture based and built on a belief that objective values exist. The search for ultimate values opens us to a higher culture. Seeking God, sanctity, and transcendence takes place in religion. Special significance is attributed to religious cults focused on the beauty of religious rituals, art, and music. The search for truth takes place most of all at university, but also in all areas of public activity. The search for goodness is manifested in ethics, whereas beauty is expressed in art. Let us add a few words about justice. If justice is to be attained, politics must be directed by truth and

goodness, and in particular, by revealed truth concerning the requirements of goodness in a particular social and political situation, that is the goodness of the community.

The search for the ultimate requires interpretation. The discovery of the deeper sense, crossing what is directly visible and transient, requires interpretation. The culture of interpretation rejects two errors of reason: fundamentalist fanaticism and subjective arbitrariness. Fundamentalism stops on the surface, on the letters. Subjectivism and arbitrariness trust in the power of a single mind. This is well illustrated by the discovery of the biblical sense. The Bible, that is the unity of the Bible books and the inspiration, or divine origin, reveals the sense. The meaning of the Bible can only be discovered within the community in which the Bible is treated as the presence of the transcendent Word. Pathologies of reason appear when it double locks itself. Firstly, it locks itself when it stops at the surface, at the letters and renounces the search for the ultimate. It then becomes the fundamentalist reason. Secondly, it locks itself as a self-sufficient monad. It breaks social bonds or considers them useless, wrongly considering itself the ultimate criterion of sense. Then, subjectivism and arbitrariness are unavoidable.

The culture of Christian humanism highly appreciates the work ethos, shaping history by cooperating with the Creator. A man is not an almighty demiurge, owner of land, nature, and the world. He is rather a gardener who was entrusted with the world. He discovers the rules of nature, respects them in his actions. He is by no means an explorer, but a gardener who develops what he was given for future generations. He cares for the condition of the environment in which he worked. He treats the Earth as our communal house.

The question about God is always present in the culture of humanism open to transcendence. The question does not determine whether God exists or not. It considers the search for God as the Unknown-Known, as a reasonable cause, that God possibly exists. It is a culture that accepts the question about God, thus being a culture directed by positive agnosticism. Think and live as if God existed. The source of agnosticism can be found in a rational conviction that the genesis of the world cannot be found in irrationality and absence of reason, but on the contrary, in Reason that creates. It believes that the beginning was not pure coincidence, but freedom of Reason. The question about God transforms into a dialog with God. It is rational to believe that the creating Reason may reveal itself, and if it reveals itself, there should be a readiness to listen to It, to accept God. The question about God and dialog with God spreads positive treatment of religion. Religious institutions preserve and contribute the necessary ethical sense to community life. Without ethics, and thus without great religious traditions, it is impossible for societies to live well or for an economy to preserve human traits or for politics to pursue a common goodness. This culture does not exclude the possibility that religions, apart from bringing ethics into the public sphere, offer a place where we can meet the creating Reason that reveals itself, the Reason that is also Love and Mercy. This possibility opens public reason to dialog with a being personifying Transcendence. Benedict XVI is convinced that the value of the culture of humanism open to transcendence is permanent: "Ce qui a fondé la culture de l'Europe, la recherche de Dieu et la disponibilité à L'écouter, demeure aujourd'hui encore le fondement de toute culture veritable" ("What was the foundation of the European culture, the search for God and readiness to listen to Him, still remains the foundation of every true culture") (Benedict XVI 2017, p. 122). Benedict's approach invites liberal atheist thought (Dworkin 2014; Lilla 2009; Gray 2020) to candid discussion and dialog. Such a dialog was provided by Jurgen Habermas, one of the important secular scholars who offers a critique of the enlightenment project (Habermas and Ratzinger 2005). Habermas's position is that secular and religious public opinion need to learn mutually to understand better moral senses (Habermas 2005).

## 5. Concluding Remarks

We can very clearly draw the border between two separate cultures. One could metaphorically describe them as two continents drifting away. One culture of the western civilizations, a bit tired, a bit too familiar, loses its appeal, but it is still alive and has

the power to attract. The other, the culture of radical enlightenment, breaks ties with everything that was before, and even when it assimilates some old concepts, it completely changes their meaning. The contemporary West is composed of these two cultures. On one hand, these cultures co-exist, but on the other hand, they compete with each other. Simultaneously, we can observe the co-existence and competition of two concepts of a man, two different political anthropologies, two pictures of human dignity, two opposing approaches to rationality and reason, two concepts of freedom and its relationship with other values, such as truth, goodness, and beauty, two contradictory views in place of God and religion in public life. Thus, we can make a list of fundamental contentious issues mentioned above.

At the present stage, we are witnessing not only a violent and fundamental dispute and rivalry over primacy, but a specific war aimed at occupying the whole public sphere. The relationship between these two cultures can be aptly described as "the clash", according to the meaning attributed by Samuel Huntington.

Pope Benedict XVI believes that both cultures should learn from each other. They have a lot in common. Similarities can especially be found between the culture open to transcendence and the culture of the first stage of enlightenment; they meet on the ground of anthropology. It was only the second, radical stage of enlightenment that made these two cultures go their separate ways. Benedict XVI points to a solution, a way out of this deadlock. Radical enlightenment should acknowledge the limitation of its rationality and be open to a broader understanding, which was common at the beginning of the enlightenment. On the other hand, the culture of contemporary western civilization should accept valuable elements offered by the rationality of radical enlightenment. According to Benedict XVI, this will enable new conversation and dialog between both cultures.

**Funding:** This research received no external funding.

**Institutional Review Board Statement:** Not applicable.

**Informed Consent Statement:** Not applicable.

**Data Availability Statement:** Not applicable.

**Conflicts of Interest:** The author declares no conflict of interest.

## Notes

[1]    The most comprehensive characteristics of the culture of radical enlightenment can be found in the speech of Cardinal Joseph Ratzinger in Subiaco on 1 April 2005. Less than three weeks later he was elected Pope, therefore, this speech must be treated as a fully personal opinion of Benedict XVI.

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
