# Peer review of "The Clash of Cultures of Radical Enlightenment and Humanism Open to Transcendence. The Perspective of Pope Benedict XVI"

_religions, doi:10.3390/rel12070460_

Round 1
Reviewer 1 Report
I thank you for your paper which I enjoyed very much. I think it is an important argument that needs to be made, and especially because it is an attempt to create a conversation between the (largely secular) social sciences and (Catholic) theology.
However, I think that this paper requires more balance, as of now it really reads as a position paper, a manifesto of a cause, rather than an engagement with a wider body of scholarship. Most of the persons cited are already those offering a Catholic critique of the Enlightenment project. It would be useful - if not essential - to refer to secular social scientists who also offer a critique of the Enlightenment project. In this way, I believe that the value of the argument will only increase.
I also felt that the potential of discussing Huntingdon's thesis within the frames you propose (which is brilliant!) was not really met.
So, all in all, I urge you to rework the essay to engage with secular scholarship that offers a critique of the Enlightenment project. I believe that this will hugely improve an already interesting and useful argument.
Author Response
- Thank you for your pertinent and convincing remarks. I agree with them. The paper was complemented by the position of Jurgen Habermas, one of the important secular scholars that offers a unique critique of the Enlightenment project. His approach aligns with Benedict XVI`s, that the Enlightenment has two phases, non-radical and radical. The former is open to dialog.
Reviewer 2 Report
This is an unclear paper. Clearly, the author struggles with expressing him/herself fluently in English and the result is a paper with much garbled and unclear language. This matters because if a reader cannot clearly grasp what an author is trying to say then much of the argument gets lost.
The paper focuses on the thoughts and ideas of a former Pope, Benedict XVI. He left office in 2013, nearly a decade ago. The paper refers to Benedict XVI as the pope, with no apparent awareness that the current pope, is Francis.
The paper alludes to the well-known arguments of Samuel Huntington. The paper refers to the 'clash of cultures' in this context. Huntington refers to the 'clash of civilisations' so it is not clear what comparison the author is trying to make, given that a different analytical term is udes compared to that of Huntington.
The paper is descriptive and much of it seems to be derived from the author's own thoughts on the issue that s/he is examining.
Author Response
- Thank you for your pertinent and convincing remarks. The paper is fully revised from a language perspective. Information that Pope Benedict XVI was the former pope and that Pope Francis is the current pope was added. It was clearly defined in the paper that the meaning of “civilization” coincides with that of “culture,” that is, conformity with European tradition of thought. In my opinion, the paper is not only descriptive but rather interpretative of the Pope`s thoughts.
Reviewer 3 Report
The author provides literature also in Polish. Therefore, he must know this language. It seems to me that one should quote original statements and not translations. Benedict XVI did not speak Polish.
Author Response
- Thank you for your pertinent remark. In the paper was added the original quote from Benedict XVI`s speech in Paris in 2008.
Reviewer 4 Report
In very interesting and important article Author presents the main element of the culture of radical enlightenment (pp. 2-6, and its aporias: pp. 6-7), such as “human dignity, political anthropology, freedom, rationality of reason, the role of religion and legitimacy of the question about God” (p. 2). His/her presentations are bona fide and honest, he/she understands the main problems of contemporary Western culture and Ratzinger’s positions toward them. Reflections on “new morality” and “political moralism” which are connected with the dominant positivistic perspective (connected even with Hens Kelsen’s juridical investigations) are very good based (although not on crucial Pope’s speeches and documents, but above all on reports of Polish and non-Polish authors). In this part we have important analyse of human dignity (but we have no information on Kantian position in this area, which is crucial for proponents of the radical enlightenment, and critical for traditional Christian orientation; I suggest introduce here some information on Kant’s position and some information on three false exegesis in 20th century on Jesus and His relations to Church which were important for Christians in Ratzinger’s book La Chiesa…: there we have important information on moral merely dimension of Christianity, one between another), human freedom (with discussion on universal culture and relativistic position and on relation between will of human person and cultural/moral rules), religion (radical enlightenment originated within the culture rooted in Christianity, but breaks all its ties with it and is directed by negative agnosticism), domination of political morality or utopia over dignity of an individual (above all utilitarian problem) and positivistic rationalism as an specific ideology (with very important notes on two opposite perspectives of rationality). In the second part of an article (number 4.) an Author presents Ratzinger’s vision of the culture of humanism open to transcendence as universal one, not (or not only) as “Christian culture”, but rooted in monotheistic premise (I suggest to use the International Theological Commission’s 2009 document In Search of a Universal Ethic: A New Look at the Natural Law, in which we have investigations on human rights founded on the innate dignity of every human person created by God; from my perspective, it is precisely this change, already mentioned by Pope Benedict XVI in his famous 2011 Bundestag speech on the increasingly counter-cultural nature of the Christian science of law; this change is determined by the context of liberal democracy and might be seen as a reaction of the Catholic Church concerned about the condition of Western societies, where liberal justifications are being used for transgressing the boundaries once set by the norms of natural law). In “Concluding Remarks” an Author once again lead the crucial points of two separate cultures which “the clash” in for him/her obvious although he/she writes on Pope Benedict XVI believe that both not only could but even should learn from each other (p. 9).
I have also more general questions: is „the Western culture” (Roszkowski 2019) at the same time „humanism open to transcendence”, as wrote an Author on p. 1? Maybe „the culture of radical enlightenment” is also the part – and now dominant part - of the Western culture? Maybe inside the Western culture we have now tension between two projects (we are reading on p. 2, that “Benedict XVI sees the clash between two dominant cultures of the west”), which are connected also with two kinds of thinking on human rights and even on (liberal) democracy? First, dominant now, is founded on „the culture of radical enlightenment”, and second – closer to Ratzinger’s thinking, but at the same time even on the “counter-cultural” position (as he said in German parliament in 2011) – is rooted in „humanism open to transcendence”?
Author wrote on p. 1: „Pope’s approach differs not from from political theology, both from the theological (…) and political philosophy point of view (…)”. Is political philosophy part of political theology? And – as he wrote below – Pope’s approach „comes close to political philosophy” or is critical to it because for contemporary political philosophers – as Habermas – religion is important element of the public life but only as moral project important for society and co-operation between various individuals?
In footnote 1. on p. 2 lack the year of the Ratzinger’s speech in Subiaco.
Author Response
- Thank you for your pertinent and convincing remarks. The paper was complemented by Kant`s position on human dignity. My interpretation of the Pope`s thoughts led to the thesis that “investigations on human rights founded on the innate dignity of every human person created by God” concerns the natural order that in consequence is has not Christian character but human.
Benedict XVI`s position is much closer to Roszkowski and Scruton’s understanding of Western Culture. The culture of radical enlightenment breaks with the former culture. Of course, this is a normative rather than a prescriptive approach.
The statement ”Pope’s approach differs not from political theology, both from the theological (…) and political philosophy point of view (…)” concerns typology, rather than the meaning of political theology, not political philosophy.
In footnote 1. on p. 2 the year of Ratzinger’s speech in Subiaco was added.
Reviewer 5 Report
I haven't comment.
Author Response
- I don`t have any comment either.
Round 2
Reviewer 2 Report
I am afraid that my initial opinion of the article has not changed. It is poorly written and rather unclear and still addresses the thoughts of Pope Benedict as though he is still the pope. There is a brief mention of Francis but no attempt to bring him into the narrative. What does it matter what Benedict thought about Huntington's 'clash of civilisations' - or 'cultures' as the author insists on addressing the issue (either though civilisaitons and cultures are not the same thing, as a little research on the topic would quickly make clear). Why is this still aburning issue, one worthy of our attention?
There is no real attempt to deal with the issues I raised in the first round of reviews. The changes to the paper are cosmetic at best.
Author Response
The issue ‘clash of cultures’ is important, even fundamental for contemporary social and political life and still is a topical item. In consequence its significance improve.
There is two traditions that interpretating relation between words “cultur’ and ‘civilisation’. First, they have different meaning. Second, they have similar meaning.
The Pope Benedict is eminent intellectualist in xx/xxi century, not only in catholic world.
The thought of Pope Benedict is important and newsworthy source at the moment to be detailed analysis and interpreting to better understand this fundamental issue ‘clash of cultures’.